# Microbiological, Physicochemical and Nutritional Properties of Fresh Cow Milk Treated with Industrial High-Pressure Processing (HPP) during Storage

**DOI:** 10.3390/foods12030592

**Published:** 2023-01-31

**Authors:** Shu Huey Lim, Nyuk Ling Chin, Alifdalino Sulaiman, Cheow Hwang Tay, Tak Hiong Wong

**Affiliations:** 1Department of Process and Food Engineering, Faculty of Engineering, Universiti Putra Malaysia (UPM), Serdang 43400, Selangor, Malaysia; 2F&N Global Marketing Pte. Ltd., Singapore 119958, Singapore

**Keywords:** milk processing, microbiological properties, physicochemical properties, vitamins, minerals, storage

## Abstract

The safety, shelf life, and quality of fresh cow milk treated using industrial High-Pressure Processing (HPP) treatment at 600 MPa for 10 min was studied to identify the novelty of this non-thermal technology in milk processing. Changes in microbiological and physicochemical properties, including nutritional values of vitamins and amino acid profiles, were measured for a 60-day storage period at 6 °C +/− 1 °C. The HPP treatment produced milk that met all microbial safety requirements and exhibited a shelf life beyond 60 days in a hot and humid region. High physicochemical stability was achieved, with consistent pH and undetectable titratable acidity. The HPP treatment successfully retained all vitamins and minerals, including calcium (99.3%), phosphorus (99.4%), and magnesium (99.1%). However, the 60-day storage caused some degradation of Vitamin A (25%), B3 (91%), B5 (35%), B6 (80%), and C (85%), and minerals, including potassium (5%) and zinc (18%) when compared with fresh milk. This research has shown that the adoption of advanced treatment with HPP is very beneficial to the dairy industry in preserving milk quality in terms of its physicochemical and nutritional properties and extending its storage shelf life beyond 60 days.

## 1. Introduction

Milk is a balanced and nutritive food that is important for human health. However, the high moisture and nutrition contents of milk cause rapid microorganism proliferation and result in spoilage. The main microorganisms present in fresh milk include *Staphylococcus aureus, Salmonella spp., Listeria monocytogenes, Escherichia coli* O157:H7, and *Campylobacter* [1], which could pose a severe hazard to humans if they unknowingly consume contaminated milk. Thermal treatments, including low-temperature long-time pasteurisation (LTLT, 63 °C/30 min), high-temperature short-time pasteurisation (HTST, 72–75 °C for 15–20 s), ultra-pasteurisation/extended shelf-life treatment (ESL, 125–128 °C for 2–4 s), and ultra-high temperature treatment (UHT, 135–140 °C for 1–3 s) are common techniques used for extending the shelf life of milk [2]. However, common LTLT or HTST pasteurisation only provide an average of 10 days of shelf life [3], while ultra-pasteurisation or extended shelf-life (ESL) milk products may have a shelf life between 15 and 30 days [3,4]. In addition to limited shelf life, thermal treatments also have detrimental effects on milk quality, such as whey protein denaturation [5], reduction of calcium, vitamins (thiamine, B12, and C), and changes in organoleptic properties [6]. In this regard, non-thermal technologies such as high-pressure processing (HPP), pulse electric field, ultrasonication, and irradiation are innovated with aims to minimise loss of nutrients by thermal treatment while still being effective in destroying pathogenic and spoilage microorganisms in milk [7]. An investigation has demonstrated that HPP offers promising potential in processing high-quality milk [8]. It normally employs pressure of 300–600 MPa at room temperature for 2–30 min to eliminate pathogenic microorganisms and extends the shelf life of milk with minimal alteration of nutritional and sensorial attributes [9]. High isostatic pressure only disrupts the non-covalent bonds (hydrogen, ionic, and hydrophobic bonds), causing alterations in cell morphology and membrane that eventually leads to cell death and achieves microbial inactivation [10]. Hence, the nutritional and sensory properties of the products are not affected. HPP also has other advantages such as low contamination risk and is declared an environmentally friendly processing technology [10]. Although HPP offers many benefits, its use by manufacturers is still limited due to high equipment costs.

Most HPP research has been performed at lab-scale, with García–Risco et al. (1998) [11] reporting the shelf life of HPP milk (45 days) when treated at 400 MPa for 30 min using an HPP apparatus (Model 900, Eurotherm Automation, Lyon, France), while Mussa and Ramaswamy (1997) [12] reported that milk subjected to HPP at 350 MPa for 32 min using an ABB Isostatic Press (ABB Autoclave System, Autoclave Engineers, Erie, PA, USA) had a shelf-life of 12–25 days when stored at 0–10 °C. At the industrial scale, Stratakos et al. (2019) [13] observed a storage life of 28 days at a storage temperature of 4 °C when milk was treated at 600 MPa for 3 min using a 35 L commercial-scale high-pressure press. Studies on the industrial scale of HPP are still limited as scale-up of HPP treatment for milk preservation from lab-scale to pilot or industrial scale is a huge challenge due to high costs and equipment availability. Tan et al. (2020) [14] reported that fresh milk’s shelf life was extended to 22 days using an industrial-scale HPP treatment (600 MPa, 5–7 min), but the data were available for only microbiological evaluations. To date, no complete study has been found reporting all microbiological, physicochemical, and nutritional contents of vitamins and minerals of HPP-treated milk at an industrial scale. This research was conducted to evaluate the effectiveness of industrial HPP treatment of locally produced fresh milk in Malaysia in a hot and humid climate with storage issues while overcoming detrimental effects of milk quality using heat treatments.

## 2. Materials and Methods

### 2.1. Processing of Milk

Fresh cow milk was collected from the university farm, Ladang 16, Universiti Putra Malaysia. Milk was mixed evenly, cooled to 4 °C, and poured into 350 mL polyethylene terephthalate (PET) bottles. Milk was treated in a 55 L high-pressure processing unit (Hiperbaric 55; Hiperbaric High-Pressure Technologies, Burgos, Spain) at 600 MPa for 10 min at 10 ± 2 °C using water as the transmitting fluid. Fresh milk was used as a control. High-pressure treated milk was labelled as HPP milk, and untreated milk as fresh milk. All milk samples were stored at 6 ± 1 °C for storage analysis. Samples were analysed for microbiological and physicochemical properties and vitamin and mineral contents from day 0. Fresh milk was kept for analysis until day 6, while treated milk until day 60. For fresh milk, microbiology analysis was conducted daily, with a 3-day interval for physicochemical, vitamin, and mineral analysis. The HPP milk samples were analysed at intervals of 5 days for microbiological, 6 days for physicochemical analysis, and 15 days for vitamin and mineral analysis during the 60-day storage.

### 2.2. Microbiological Analysis

Total plate count (TPC) and total yeast and mold count in milk were determined using procedures described in Chapters 3 and 18 of the FDA Bacteriological Analytical Manual (BAM) [15]. The total coliform count was calculated using the AOAC Official Method 991.14 by incubating for 24 ± 2 h at 35 ± 1 °C; then, the coliforms appeared as red colonies with one or more gas bubbles counted promptly using the standard colony counter [15]. Mesophilic, thermophilic aerobic spore count, and psychrotrophic bacteria count were determined following Chapters 23, 26, and 13, respectively, in the *Compendium of Methods for the Microbiological Examination of Foods* (CMMEF) [16] with different incubating times and temperatures, where mesophilic incubated anaerobically at 35 °C for 3 to 5 days, thermophilic aerobic at 55 ± 2 °C for 72 h while psychrotrophic bacteria at 17 ± 1 °C for 16 h, followed by 3 more days at 7 ± 1 °C. Psychrotrophic spore count was determined. *Bacillus cereus, Staphylococcus aureus,* and *Listeria monocytogenes* were determined following FDA BAM Chapters 14, 12, and 10, respectively [15]. *Escherichia coli (E. coli)* and *Clostridium perfringens* were determined by AOAC Official Method 991.14 [15] and 976.30 [15], respectively. For *E. coli*, the plates were incubated for 48 ± 4 h at 35 ± 1 °C. After the incubation period, *E. coli* colonies appeared as blue colonies associated with gas bubbles and were counted promptly. For *Clostridium perfringens,* tryptose sulfite cycloserine (TSC) agar was used for the incubation of black colony for 20 h at 35 °C. *Salmonella* spp. was determined using the enzyme-linked immunosorbent assay (ELISA) method as described by Veling et al. [17]. Colonies were enumerated, and results were expressed as the logarithm of colony-forming units (log_10_CFU/mL). Colony-forming unit (CFU) is a unit used in microbiology to measure the number of viable microorganisms (bacteria, fungi, viruses, etc.) in a sample that can reproduce through binary fission under controlled conditions.

### 2.3. Physicochemical Analysis

The pH of cow milk samples was measured using a pH meter (Mi805, Milwaukee, Hungary). Titratable acidity (TA) was determined according to the Association of Official Analytical Chemists (AOAC) Official Method 947.05 using phenolphthalein indicator to titrate with 0.1M NaOH and expressed as a percentage of lactic acid. Specific gravity was determined following the method for the specific gravity of water in AOAC Official Method 955.37, using a pycnometer. Total protein was determined using formaldehyde titration described by Pyne’s method [18], in which oxalate was used as an alkali for the titration, and total protein was calculated using a factor of 1.74. The total solid of cow milk was determined using the oven drying method at 102 ± 2 °C for 2.5 h in accordance with AOAC Official Method 16.032 [19]. Fat content was determined using the Roese–Gottlieb Method in AOAC Official Method 905.02, which involved extraction, distillation, and drying. Solid-non-fat (SNF) was determined by subtracting the total solid from the total fat content.

### 2.4. Nutritional Analysis

#### 2.4.1. Vitamin Content Analysis

Vitamin A (Retinol) and E (alpha-tocopherol) were determined using high-performance liquid chromatographic methods (HPLC) after alkaline saponification. Vitamin B1, B2, and B6 were analysed following the method described by Agostini-Costa et al. [20]. Vitamin B5 was determined based on Food Chemistry (2000), similar to the method reported by Woollard et al. [21]. Vitamin B7, B9, and B12 were analysed based on AOAC Official Method 960.46 using a microtiter assay [22]. In this test, the test vitamin extract was pipetted to the walls of a microtiter plate coated with microorganisms, followed by incubation at 37 °C for 44–48 h until all vitamins were consumed. After the incubation period, the absorbances were measured using a microtiter plate reader at 610–630 nm. Vitamin C (ascorbic acid) was determined according to AOAC 967.21 using 2,6-dichlorophenolindophenol (DCPIP) titration [23]. Vitamin K was determined based on AOAC Official Method 999.15, where extraction was performed using hexene, followed by HPLC determination [24]. Vitamin D was determined based on AOAC Official Method 995.05 via a single liquid–liquid extraction followed by saponification, solid-phase extraction, and evaporation, as described by Silva and Sanders [25].

#### 2.4.2. Mineral Content Analysis

Milk mineral content, including Calcium (Ca), Potassium (K), Magnesium (Mg), Phosphorus (P), Zinc (Zn), and Selenium (Se), first underwent acid digestion as reported by Kira and Maihara [26], followed by mineral content analysis based on AOAC Official Method 984.27 (final action 1986) using inductively coupled plasma optical emission spectrometry, ICP-OES (Varian 720-ES, Varian Inc., Walnut Creek, CA, USA) with a solution of yttrium as internal standard.

### 2.5. Statistical Analysis

All measurements were performed in triplicate with samples prepared from the same batch of milk, except for logistic reasons, Vitamin K was without repetition. Mean values were compared using analysis of variance (ANOVA) using the tool XLSTAT by Addinsoft and Microsoft Excel software. Tukey’s test was performed to compare the differences between the mean values at a confidence level of 0.05.

## 3. Results and Discussion

### 3.1. Microbiological Analysis

HPP treatment successfully decreased TPC by 75% and other bacteria counts in milk by 100% (Table 1), meeting food safety limits and local industrial standards [27]. Figure 1 shows that storage of HPP-treated milk for 60 days gave high microbiological stability as its microbial activities were under control when compared to the definite unsafe condition of fresh milk with the continuous growth of all microbiological bacteria stored at 6 ± 1 °C for its first 6 days (Appendix A).

Total Plate Count (TPC) is a method of estimating the total number of microorganisms in products commonly used by dairy manufacturers for determining the microbiological quality of milk. In the Malaysian Food Act [27], the limit of TPC is ≤10^5^ CFU/mL for safe consumption, but the TPC of fresh milk was recorded at 5.13 log_10_CFU/mL, and it increased significantly (*p* < 0.05) to 6.19 log_10_CFU/mL after 6 days of storage while HPP treated milk was successfully reduced to 1.26 log_10_CFU/mL and maintained below 2 log_10_CFU/mL for 60 days of storage (Figure 1a), meeting the safety limit (Table 1). A lab-scale HPP study by Liepa et al. [28] reported a pronounced reduction in TPC by 99.7% using HPP (STANSTED fluid power LTD, Stansted, Harlow, UK) at 550 MPa for 3 min, while Razali et al. [29] reported HPP (Avure 2L-700 HPP Laboratory Food Processing System, Avure, Kent, Washington, DC, USA) at 400 Mpa for 5 min reduced TPC to an undetectable level in milk. In a similar study using the same commercial unit (Hiperbaric 55, Hiperbaric, Spain), Tan et al. [14] reported a high TPC reduction of 99.98% at 600 Mpa for 5–7 min. Stratakos et al. [13] showed that commercial scale HPP (Quintus 35 L, Avure Technologies, Kent, Washington, DC, USA) at 600 Mpa for 3 min reduced TPC by 83.33% and prolonged the shelf life of milk to 28 days.

Yeasts and mould can cause the development of off-flavours in milk due to the generation of toxic metabolites by mycotoxins which restrict the shelf life of milk and pose a potential health risk [30]. Fresh milk used in this study had yeast and molds count above the safety limit of < 1 log_10_CFU/mL [30]. HPP treatment reduced the count to safe and undetectable levels (Figure 1b). A similar result was reported by Tan et al. [14], where most vegetative yeasts and moulds in cow and goat milk were inactivated within 5–7 min at 450–600 Mpa. The effectiveness of HPP in controlling yeast and mould count in this study has shown stability at an undetectable level throughout the 60 days of storage (Figure 1b) which is longer than studies by Tan et al. [14] for 22 days.

Spore-forming bacteria, which include the psychrotrophic, mesophilic, and thermophilic spore formers, are common in dairy products due to their ability to survive under different temperatures [31]. Psychrotrophic microorganisms constitute a major cause of milk spoilage [32] due to their ability to produce heat-resistant enzymes such as proteases, lipases, and phospholipases under refrigeration [33,34]. The psychrotrophic bacteria count of high-quality fresh milk should be <10^5^ CFU/mL based on the Malaysian Food Act [27], but this study shows that the psychrotrophic bacteria count in fresh milk was higher at 6.81 ± 0.05 log_10_ CFU/mL (Table 1). The high psychrotrophic bacteria was reduced to an undetectable level after HPP treatment, as were its spores and both the mesophilic and thermophilic aerobic spores. This result is supported by Tan et al. [14], who reported that psychrotrophic and mesophilic spores were not detected in HPP milk (600 Mpa, 7 min). The psychrotrophic bacteria count of HPP milk was also not affected by long refrigeration storage, which was maintained at undetectable levels throughout the 60 days of storage (Figure 1c).

Other microbiological properties with counts that were all well-kept below the safety limits during the 60 days of storage after HPP treatment, although they had counts initially exceeding safety levels before treatment, are the coliform (1.7 log_10_CFU/mL) and *Staphylococcus aureus* (1.3 log_10_CFU/m). This suggests that HPP treatment had high efficiency in ensuring treated milk is safe despite unavoidable conditions in the milking process and farms causing microbial contamination. Several researchers who have reported similar findings on the effectiveness of HPP treatment in reducing these bacterial counts include Stratakos et al. [13] on *E. coli* reduction by 5.6 and 6.8 log_10_CFU/mL at 600 Mpa for 3 and 5 min, respectively, Liu et al. [35] on *E. coli* reduction by 2.9 log_10_CFU/mL at 600 Mpa for 5 min, and Tan et al. [14] on total coliform reduction by 1.6 log_10_CFU/mL at 450–600 Mpa for 5–7 min.

Pathogenic bacteria such as *Bacillus cereus, Clostridium perfringens, Listeria monocytogenes,* and *Salmonella spp.* were all undetected in fresh milk and also throughout the storage for HPP milk. For safety consumption, *Bacillus cereus* and *Clostridium perfringens* in food should be <10^5^ CFU/mL [36] and <10^4^ CFU/g [37], respectively, while *Listeria monocytogenes* and *Salmonella spp.* should not be detected in a 25g sample [38].

### 3.2. Physicochemical Analysis

Figure 2 shows slight changes in all physicochemical properties of HPP milk during 60 days of storage except for acidity, which was reduced from initially 0.20% to <0.1% after HPP treatment (Figure 2b). Acidity reduction accompanied by pH value increase after HPP treatment at 600 MPa was also reported in other studies [15,29]. During HPP, the casein micelle disaggregation alters minerals distribution and raises the concentration of ionic calcium in milk, resulting in an increase in the phosphate concentration of the milk serum and pH [28].

HPP milk had high pH stability throughout 60 days of storage, indicating it has low bacteria load to produce lactic acid and generate free fatty acid through fat lipolysis [39]. Tan et al. similarly reported a pH decrease in HPP milk stored for 22 days [14]. It is common to have an acidity increase in spoilt milk, as observed in fresh milk with 6 days of storage (Appendix B), normally associated with an increase in free fatty acids [40], age gelation [41], or browning reactions [42] during storage.

HPP did not cause many changes to the protein and fat contents of milk during its 60 days of storage (Figure 2c,d). This result was consistent with Tan et al. [14], who reported no significant changes (*p* > 0.05) in the total protein and fat content of HPP milk samples during 22 days of storage. High pressure will only influence the bonds which stabilise the spatial structure of proteins and cause reversible or non-reversible denaturation depending on the pressure level [43]. Most studies have proven the effectiveness of HPP in preserving milk fat, as high pressure does not damage milk fat globule membranes; thus, lipolysis is prevented, and fat content is retained [44,45].

HPP treatment was also found to cause no significant changes (*p* > 0.05) to the total solids (Figure 2e) and non-fat milk solids (Figure 2f), similar to those reported by Tan et al. [14]. However, these contents changed significantly (*p* < 0.05) in an inconsistent manner during 60 days of storage. The increase in total solids and non-fat solids of HPP-treated milk could be due to the precipitation fraction of large casein–casein and the formation of casein–fat aggregates during HPP [46] or induced fat crystallisation due to the duration of pressure treatment and storage [47]. Specific gravity was not affected throughout this study; it was retained at 1.02–1.03 g/mL.

### 3.3. Nutritional Analysis

#### 3.3.1. Vitamin Content

Most of the vitamin contents of milk in the present study, which includes fresh milk, were found in relatively low levels compared to commercially fortified milk, probably due to the nature of milk in which vitamins are not present as a primary source of milk. Some of these vitamins, which include beta-carotene and Vitamin A, B1, B2, B3, B6, C, and E, were too low and were below the limit of reporting (LOR) or below the minimum concentration of a substance in a sample that can be reliably detected by a laboratory, hence levels below this limit can cause variation and affect accuracy. Despite the low vitamin levels, HPP treatment did not cause significant changes (*p* > 0.05) in milk vitamins except for the slight Vitamin C increase (Table 2). This was unexpected and can be explained by the variations due to the detection level below LOR. Sierra and Vidal–Valverde [48] found no significant (*p* > 0.05) losses of Vitamin B1 and B6 in whole milk after HPP (400 MPa, 30 min), while Moltó-Puigmartí et al. [49] also reported no loss in the total Vitamin C level in human milk after HPP (600 MPa, 5 min, 22–27 °C). Retention of vitamins in milk was probably due to the property of vitamins, which consists of small molecules and covalent bonds that were not affected by high pressure [50]. The storage study here shows that vitamins in HPP milk reduced gradually over the 60-day storage (Figure 3). Most vitamins in HPP milk, including Vitamin A (25%), B3 (91%), B5 (35%), B6 (80%), and C (85%), fell below their original level at the end of storage, except for Vitamin B7 (25%), B9 (100%), and B12 (20%) with increment. There is no comparison of other work on vitamin deterioration in HPP-treated milk during storage, but records of vitamin deterioration in fresh milk during 6 days of refrigeration storage were found for Vitamin A, B6, B12, and C (Appendix C). It is similarly reported that although vitamins were well-retained in juices after high-pressure treatment, they were also degraded during refrigeration storage [51,52].

#### 3.3.2. Mineral Content

The milk samples used in this study had the highest mineral content of potassium, followed by calcium, phosphorus, magnesium, and zinc (Figure 4). There was no significant difference (*p* > 0.05) in the mineral contents of milk after HPP treatment (Day 0). It successfully retained calcium, phosphorus, and magnesium contents by 99.3%, 99.4%, and 99.1%, respectively. However, significant changes (*p* < 0.05) were observed in the mineral contents of HPP milk over the 60 days of storage in an inconsistent manner (Figure 5). Significant loss (*p* < 0.05) was observed in potassium (5.3%) and zinc (18.4%), whereas calcium, phosphorus, and magnesium contents increased by 1.6%, 1.1%, and 13.1%, respectively, after storage when compared with the original fresh milk. Selenium remained undetected at levels <0.2 mg/kg throughout this study. Andrés et al. [53] observed no significant (*p* > 0.05) changes in mineral profiles of potassium, calcium, magnesium, and zinc in milk after HPP (450–650 MPa, 3 min, 20 °C) during the 45 days of storage at 4 °C. The inconsistent changes were more prominent for fresh milk kept refrigerated for 6 days of storage, with a higher degradation of potassium (4.3%), calcium (7.2%), and phosphorus (14.8%), except for selenium which also remained at undetectable level (Appendix D).

## 4. Conclusions

This research has shown the potential of HPP treatment in preserving milk quality for the dairy industry. This novel technology gives promising results in terms of not only product safety and nutritional properties but also extending shelf life significantly when compared to conventionally heat-treated milk. HPP-treated milk had a storage shelf life beyond 60 days, with all microbial testing meeting permitted safety levels. It had high stability for physicochemical properties with consistent pH and acidity during the entire storage. HPP treatment has successfully retained calcium, phosphorus, magnesium, and zinc contents by 99.3, 99.4, 99.1, and 100%, respectively. The HPP treatment itself did not cause much vitamin and mineral deterioration. However, some changes were observed for the vitamin and mineral contents at the end of 60-day storage. Degradation was observed for Vitamin A, B3, B5, B6, and C, and the minerals potassium and zinc, while an increase in Vitamin B7, B9, B12, calcium, phosphorus, and magnesium contents were detected. Future HPP research focusing on milk vitamins and minerals to overcome this study limitation is suggested.

## Figures and Tables

**Figure 1 foods-12-00592-f001:**
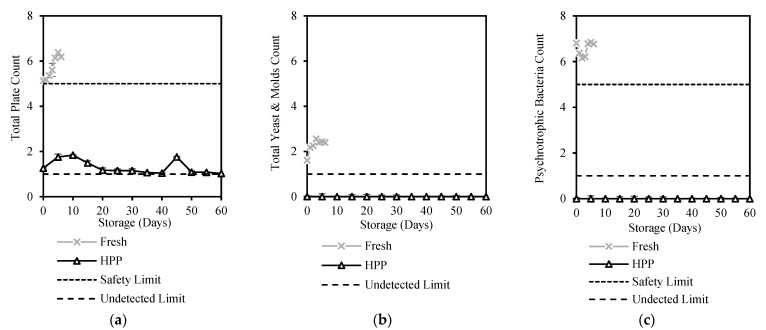
Changes in (**a**) Total Plate Count, (**b**) Total Yeast and Molds Count, and (**c**) Psychrotrophic Bacteria Count of HPP and ESL milk during 60-day storage (log10 CFU/mL).

**Figure 2 foods-12-00592-f002:**
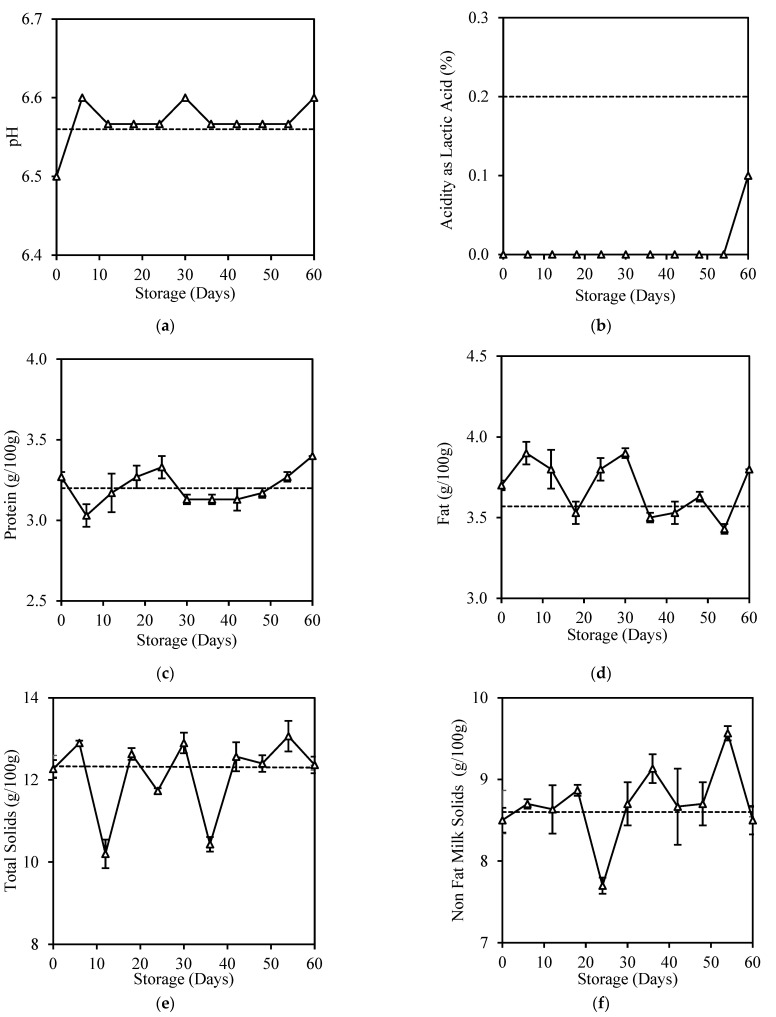
Changes in (**a**) pH, (**b**) Acidity as Lactic Acid, (**c**) Protein, (**d**) Fat, (**e**) Total Solid, and (**f**) Non-Fat Milk Solids of HPP milk during 60-day storage. Dotted lines represent the fresh milk before treatment.

**Figure 3 foods-12-00592-f003:**
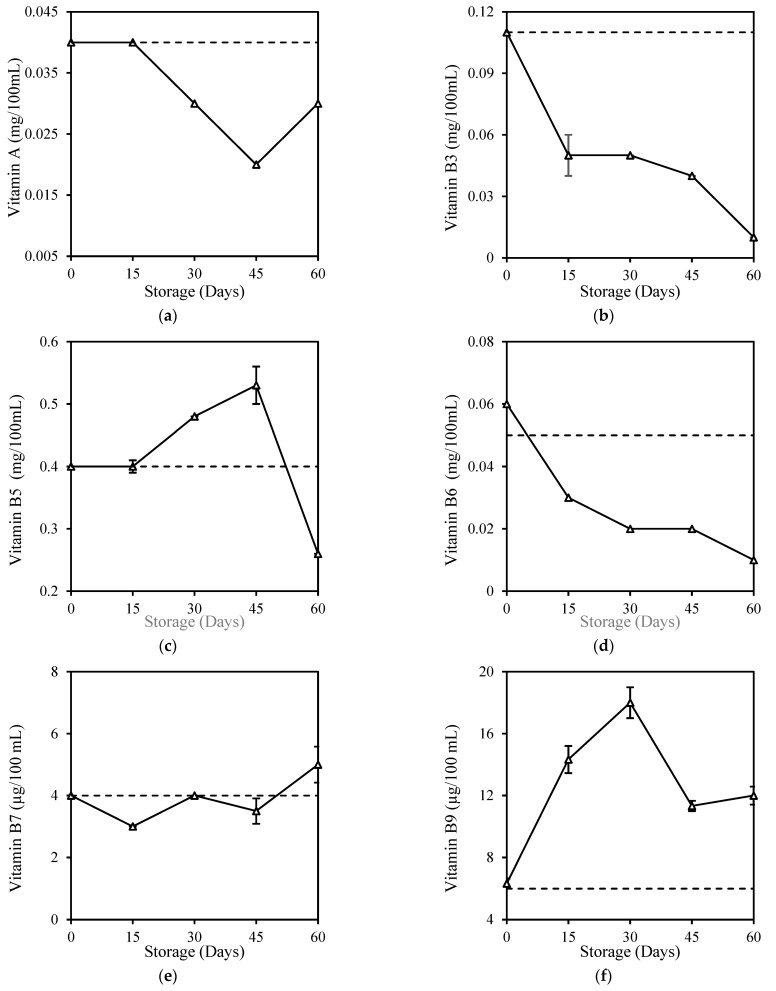
Changes in (**a**) Vitamin A, (**b**) Vitamin B3, (**c**) Vitamin B5, (**d**) Vitamin B6, (**e**) Vitamin B7, (**f**) Vitamin B9, (**g**) Vitamin B12, and (**h**) Vitamin C of HPP and ESL milk during 60-day storage. Dotted lines represent the fresh milk before treatment.

**Figure 4 foods-12-00592-f004:**
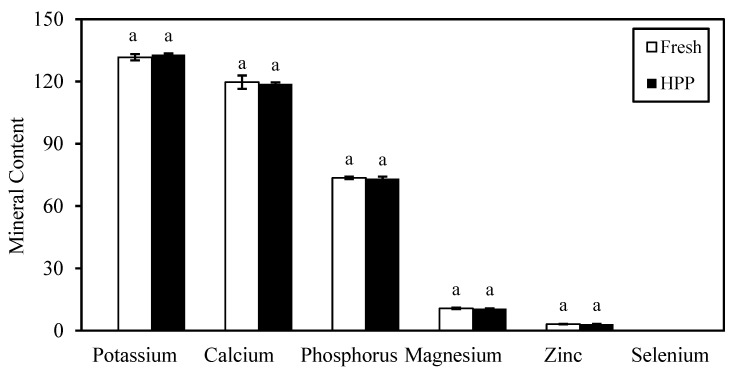
Changes in mineral content of fresh milk after HPP treatment. Potassium, Calcium, Phosphorus, and Magnesium in unit mg/100 mL; Zinc and Selenium in unit mg/kg. Alphabet “a” represent no significant difference between bar at *p* > 0.05.

**Figure 5 foods-12-00592-f005:**
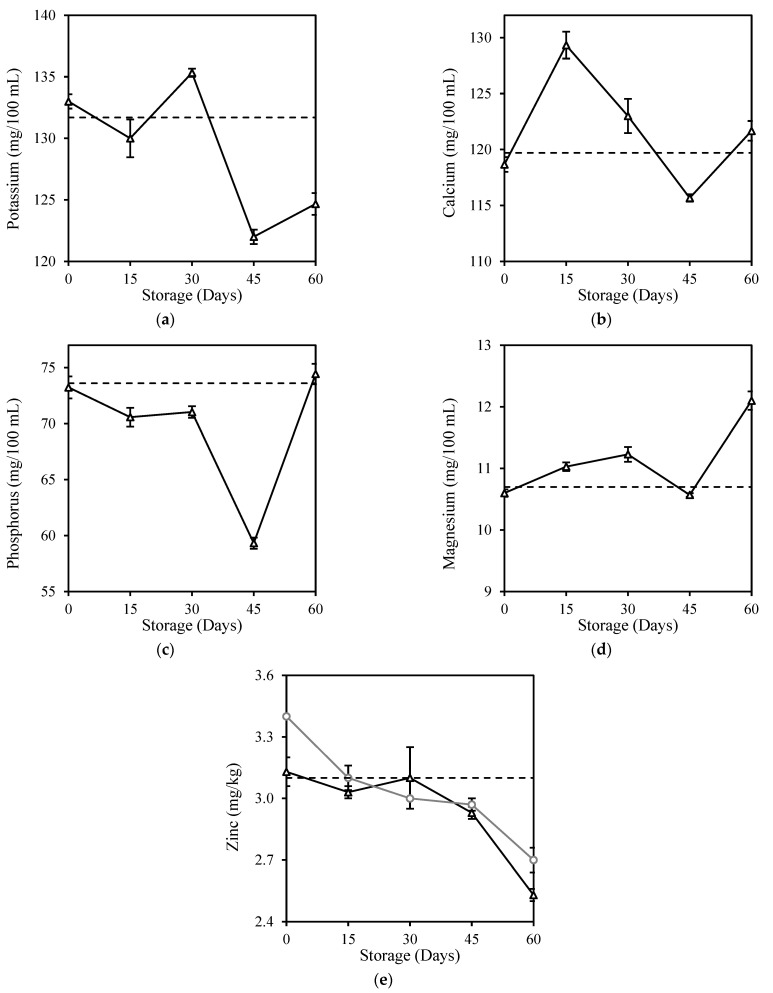
Changes in (**a**) Potassium, (**b**) Calcium, (**c**) Phosphorus, (**d**) Magnesium, and (**e**) Zinc contents of HPP and ESL milk during 60-day storage. Dotted lines represent the fresh milk before treatment.

**Table 1 foods-12-00592-t001:** Effects of HPP on microbiological properties of fresh milk.

Bacteria Count (log_10_ CFU/mL)	Fresh Milk	HPP	Safety Limit
Total Plate	5.13 ± 0.06 ^b^	1.26 ± 0.04 ^a^	<5
Total Coliform	2.56 ± 0.02	<1	<1.7
*Escherichia coli*	1.53 ± 0.12	<1	<2
Yeast and Molds	1.61 ± 0.15	<1	<1 *
Psychrotrophic Bacteria	6.81 ± 0.05	<1	<5
Psychrotrophic Spore	0.99 ± 0.15	<1	<1 *
Mesophilic Aerobic Spore	0.72 ± 0.06	<1	<5
Thermophilic Aerobic Spore	<1	<1	<1 *
*Staphylococcus aureus*	2.96 ± 0.03	<1	<1.3
*Bacillus cereus*	<1	<1	<5
*Clostridium perfringens*	<1	<1	<4
**Bacteria Existence (Present/Absent)**	**Fresh Milk**	**HPP**	**Safety Limit**
*Listeria monocytogenes* per 25 gm	Absent	Absent	Absent
*Salmonella* spp. per 25 gm	Absent	Absent	Absent

^ab^ Mean ± standard deviation in same row with different superscripts letters are significantly different at *p* < 0.05. Safety Limit values were compiled from Laws of Malaysia P.U.(A) 437 Of 1985 Food Act 1983 Food Regulations 1985 Arrangement of Regulations, Regasa et al. (2019), Official Journal of the European Communities (1990), Gilbert et al. (2000) and Food Standards Australia New Zealand, (2018) except for * which was obtained from industry (Fraser and Neave).

**Table 2 foods-12-00592-t002:** Effects of HPP on vitamin content of fresh milk.

Vitamin Content	LOR Unit	Fresh Milk	HPP
Vitamin A (as Retinol) *	0.1 mg/100 mL	0.04 ± 0.00 ^a^	0.04 ± 0.00 ^a^
Beta-carotene *	1 mg/100 mL	0.02 ± 0.00 ^a^	0.02 ± 0.00 ^a^
Vitamin B1 *	0.25 mg/100 mL	0.04 ± 0.00 ^a^	0.03 ± 0.00 ^a^
Vitamin B2 *	0.25 mg/100 mL	0.17 ± 0.00 ^a^	0.18 ± 0.00 ^a^
Vitamin B3 (as Niacin) *	0.25 mg/100 mL	0.11 ± 0.00 ^a^	0.11 ± 0.00 ^a^
Vitamin B5	0.25 mg/100 mL	0.40 ± 0.00 ^a^	0.40 ± 0.00 ^a^
Vitamin B6 *	0.25 mg/100 mL	0.05 ± 0.01 ^a^	0.06 ± 0.00 ^a^
Vitamin B7 (Biotin)	1 µg/100 mL	4.00 ± 0.00 ^a^	4.00 ± 0.00 ^a^
Vitamin B9 (Folic Acid)	1 µg/100 mL	6.00 ± 0.00 ^a^	6.33 ± 0.33 ^a^
Vitamin B12 (Cyanocobalamin)	1 µg/100 mL	1.67 ± 0.58 ^a^	2.00 ± 0.00 ^a^
Vitamin C *	2 mg/100 mL	1.00 ± 0.00 ^a^	1.67 ± 0.33 ^ab^
Vitamin D	0.1 µg/100 mL	2.50 ± 0.00 ^a^	2.50 ± 0.00 ^a^
Vitamin E (Alpha-Tocopherol) *	0.1 mg/100 mL	0.07 ± 0.00 ^a^	0.07 ± 0.00 ^a^
Vitamin K	0.1 µg/100 g	0.3	0.65

^ab^ Mean ± standard deviation in same row with different superscripts letters are significantly different at *p* < 0.05; * Results showing vitamin content lower than LOR. LOR is the limit of reporting.

## Data Availability

The data are contained within the article.

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
