# Peer review of "Microbiological, Physicochemical and Nutritional Properties of Fresh Cow Milk Treated with Industrial High-Pressure Processing (HPP) during Storage"

_foods, 2023, doi:10.3390/foods12030592_

Round 1
Reviewer 1 Report
line 77-78- why is "day, days" capitalized?
The entire methodology section is written too generally. The authors refer to the Codex (AOAC), devote a lot of space to quoting the methodology, but I think that in order to increase the readability of the methods, they should be quite thoroughly described. With this remark, of course, I do not undermine the credibility and correctness of the methods used - only their presentation.
table 1. - deserves praise for showing not only the results for fresh milk and HHP, but also the limit values - safe
figure 3c should be corrected - max value for the others (3a 3b) is 8. fla 3c about 9
figure 3c should be corrected - max value for the others (3a 3b) is 8. fla 3c about 9
What always puzzled me and at the same time interested me (because from the microbiological aspect it is irrelevant) is what happens to a destroyed cell of microorganisms? It is obvious that it does not affect the stability, but in the light of these studies, there are no significant changes in the physicochemical composition? Not knowing the results of this work, I assumed that an increase in protein content could be observed - which is not clear from the work, my question is more of a cognitive type than questioning the result of the work. Would be grateful if the authors have such knowledge for clarification but adding it in the thesis (in the introduction after line 45-46).
Reviewer 2 Report
These are my suggestions for the paper entitled "Microbiological, Physicochemical and Nutritional Properties of Fresh Cow Milk Treated with Industrial High-Pressure Processing (HPP) during Storage":
- please, rewrite the Abstract - this part of the paper is very important, while your current version is short and not enough informative (add background, aim, etc.)
- The Introduction part is good, but I need to require a microbiological profile of milk and current trends using high pressure processing for decreasing of microbiological contamination in milk and dairy products. Authors can make some state-of-the part and add one paragraph.
- All methods in Section 2.2. need to be cited as references if the Authors indicate that they used standardized methods. Or add all necessary details for reproductivity of the selected analysis.
- line 94 rewrite Coli to coli
- line 95 spp. is not italic, please revise
- line 97 add explanation what is CFU
- Table 1 - in order to avoid repetition of word "count", separate qualitative and quantitative methods. Also delete "total" for E. coli
- Figure 2 - Author can used narrower scale for Y-axis to make better visuality of all graph
- The Conclusion must be expanded with all necessary details for the study, but also with future implication of this investigation.
-
Round 2
Reviewer 2 Report
/